# Service Evaluation of the English Refugee Health Information System: Considerations and Recommendations for Effective Resettlement

**DOI:** 10.3390/ijerph181910331

**Published:** 2021-09-30

**Authors:** Thomas James Dunn, Annabel Browne, Steven Haworth, Fatima Wurie, Ines Campos-Matos

**Affiliations:** 1Population and Lifespan Sciences, The University of Nottingham, Nottingham NG5 1PB, UK; 2Public Health England Migrant Health Team, London SE1 8UG, UK; annabel.browne@phe.gov.uk (A.B.); shawora@essex.ac.uk (S.H.); fatima.wurie@phe.gov.uk (F.W.); ines.campos-matos@phe.gov.uk (I.C.-M.); 3Institute for Social and Economic Research (ISER), University of Essex, Colchester CO4 3SQ, UK; 4Institute of Epidemiology and Health Care, University College London, Gower Street, London WC1E 6BT, UK

**Keywords:** refugees, global health, information science

## Abstract

Since 2015, the UK has resettled over 25,000 refugees. To support resettlement and integration, refugees undergo a pre-arrival medical health assessment (MHA), which is used for healthcare planning by local government in England. This study aimed to understand the utility and effectiveness of the MHA and flow of data to support resettlement planning. Seven local government representatives were interviewed regarding their experiences and perceptions of the refugee health information system (HIS) and the MHA for resettlement in England. Data was analyzed using thematic analysis. The three themes indicated that the HIS was perceived to be effective, however, issues on governance, timeliness of information and access were identified. Findings showed that for the MHA to be more useful for planning, assessments for mental health issues and child special educational needs (SEN) are needed. Findings also indicated resettlement promoted joint working and acceptability of refugee resettlement. In areas where data sharing and governance processes are well defined, the HIS is effective and the MHA supports resettlement. National agencies should put structures in place to support timely health information flow.

## 1. Introduction

At the end of 2020 around 82.4 million people had been forcibly displaced due to violence, persecution, conflict and human rights violations. An estimated 26.4 million were refugees, with 6.7 million originating from the Syrian Arab Republic [1].

Refugees are defined by the United Nations High Commissioner for Refugees (UNHCR) as “someone who is unable or unwilling to return to their country of origin owing to a well-founded fear of being persecuted for reasons of race, religion, nationality, membership of a particular social group, or political opinion” [2].

In 2019, the vast majority of displaced people that applied for resettlement in Europe were settled in Germany (9640), Sweden (5408), Norway (3507), the United Kingdom (UK) (3507), and France (3311). Fewer were settled in the southern region of Europe, such as Spain (1193), Italy (413) and Portugal (879) [3]. Since 2015, the UKhas resettled over 25,000 refugees [4], 19,768 of the most vulnerable were resettled under the Vulnerable Persons Resettlement Scheme (VPRS) [5]. Resettlement priority for the UK is based on the urgency of security and/or medical needswith additional priority to women and children that are at risk of violence or exploitation or with family links in the UK [6]. Most schemes offer Leave to Remain residency for five years, with Gateway providing Indefinite Leave to Remain [5,6]. The UNHCR provides an out of country assessment for each application to determine their eligibility and refugee status [6]. Cases eligible for the UK resettlement schemes are forwarded to the Home Office who determine if the refugee meets their criteria. For refugees accepted on a UK resettlement scheme they have a pre-departure medical health assessment (MHA). The MHA is administered by the International Organization for Migration (IOM) on behalf of the Home Office. The MHA is shared with local government through a Health Information System (HIS) process that uses a secure platform. The IOM’s MHA is similar to 80 other countries' pre-departure assessment [7]. Therefore, the findings from this study could be applicable to international refugee resettlement.

Resettlement of refugees in England is overseen by local government (‘local authority’) in conjunction with Clinical Commissioning Groups (CCG) that represent the local primary and secondary care health services. The local authority will arrange housing, school places, and adult English language lessons. The Voluntary, Community and Social Enterprise (VCSE) provide refugee support services for local authorities and CCGs. The VCSE also supports community sponsorship programs where community groups provide accommodation and resettlement support for refugees. The Home Office provides overarching support and funding to the local authorities and CCGs. Monitoring and evaluation of refugee integration is a local authority responsibility as the Home Office has little contact with the refugee post-arrival [5]. The MHA is cascaded from IOM to the local authorities as shown in the HIS in Figure 1. The refugees will also receive a copy of their MHA for their records [8].

Pre-entry health assessments allow early detection and treatment of conditions, that may have wider public health implications [7,8] and, crucially, allow the host country to prepare and provide provision of appropriate services on arrival. This information allows local authorities and health partners to assess if they can meet the needs of recipients (Table 1), and to organize care on arrival. This is of importance as refugees coming to England are, by definition, vulnerable. They may have experienced periods without access to medical care and exposure to trauma, violence, communicable diseases and other health threats.

Refugees are a diverse population, with complex and widely differing needs and experiences, both pre- and post-migration [9,10]. The importance of an effective HIS in the development of interventions and support that meets the needs of refugees has been identified [5]. Lack of collaboration between health and government organisations and the transfer and flow of health information is recognised as an obstacle for the continuity of care for refugees [11]. This can be compounded by difficulties ensuring the proper transition of medical records [12]. There is a recognition that effective HIS are essential for refugee resettlement but there is a paucity of evidence on the effectiveness of HIS and health information [13].

The study will be the first to assess the utility and effectiveness of the HIS, the MHA and the data flow cascade in England. This study aims to:Evaluate if the refugee HIS is effective for English resettlement partners to identify and share medical information to organize care and treatment upon arrival.Investigate if the use of the refugee MHA meets its stated aims (Table 1) and if it provides sufficient information for resettlement.

## 2. Materials and Methods

This study was based on a policy evaluation framework from the English Government’s Magenta Book [14]. The Magenta Book provides a comprehensive overview of evaluation for Government policy, delivery and analysis professionals. The evaluation framework allows a systematic assessment of a policy design, implementation, process and outcomes [14].

### 2.1. Interview Topic Guide

A topic guide was created with a Consultant in Public Health (I.C.-M) and a migrant health researcher (F.W.). A pilot of the topic guide was carried out with a local authority resettlement lead who was not included in the final sample. From this pilot there was a need to streamline and clarify some questions. The final product was a semi-structured topic guide with open ended questions. This gave participants an opportunity to describe their perceptions and experiences. The interview topic guide is in Appendix A.

### 2.2. Sample

A purposive sample of seven areas was selected to provide a range of local authority structures, different resettlement experiences, geographical spread and urban/rural distribution in England. All participants had experience within the last 12 months of supporting refugee resettlement.Participant information sheets and consent forms were sent before interviews. All participants gave consent to take part in the study. Interviews were at a time convenient for the interviewee. Interviews were conducted on Microsoft (MS) Teams with two interviewers present (T.J.D.), (A.B.). Recordings were saved on a secure network that was password protected in line with information governance principles to protect the confidentiality of the participants. The lead interviewer (T.J.D.) asked the questions and the second interviewer (A.B.) took field notes. The participants did not receive any compensation in exchange for being interviewed. Participants could opt out at any time and request their information destroyed.

### 2.3. Data Analysis

Themes were created through thematic analysis [15]. Interviews were transcribed verbatim by the second interviewer (A.B.) and these were cross checked by the lead interviewer (T.J.D.) against audio files and field notes for accuracy and completeness. Interview transcripts were saved in MS Word and coded in MS Excel. Transcripts and codes were deidentified to ensure confidentiality and were saved on secure government network that only the two interviewers (A.B.), (T.J.D.) could access.

The first stage of analysis was familiarization, the two interviewers read and re-read each transcript. Secondly, each interviewer separately coded each transcript to support credibility and assurance and created a code list. The code list was harmonized to create a coding framework on MS Excel and transcripts were reviewed a final time. Thirdly, initial emergent themes were generated by each interviewer and collated in MS Excel. Fourthly, a review between the three lead authors formed main and sub themes, with the two interviewers (A.B.),(T.J.D.) justifying how each emergent theme was created through coding to the third author (S.H.). Fourthly, themes were defined and named by these three authors (A.B.),(T.J.D.),(S.H.). The fifth and final phase was writing the themes, sub themes and corresponding quotes that are presented in the results. All five authors were involved in this step to increase the rigor of the theming. The coding and theming tree is provided in Appendix B. Transcripts and codes were returned to participants to allow them to validate the findings and no challenges were received.

### 2.4. Ethical Approval

Internal Public Health England ethical approval was granted in November 2020. Further ethical approval through the National Health Service (NHS) was not required as this study is defined as a service evaluation by the NHS Health Research Agency [16].

## 3. Results

### 3.1. Characteristics of the Participants

All seven resettlement partners that were approached agreed to participate. Table 2 shows their characteristics. To protect the identity of the resettlement partners, only limited demographic information can be provided.

### 3.2. The HIS MHA Flow Cascade Is Effective, However, Some Areas Have Issues

The HIS is effective in identifying and sharing medical information for most resettlement providers. There are difficulties with data sharing processes for the MHA in some areas. Several providers expressed concerns with the timeliness and relevance of the health information.

#### 3.2.1. Data Sharing Processes and Information Governance Can Be a Barrier That Joint Working Can Overcome

Two participants had issues understanding what, and how relevant MHA information could be shared from host local authorities to VCSE organisations, health partners and other local authority departments.

“I think there are limitations and confusion in our area about confidentiality and data sharing and so it doesn’t work as smoothly as it could, and it is because of the different levels of professionalism between partners and providers.” Participant 5.

Some areas have data sharing agreements or secure platforms, others are unsure of the General Data Protection Regulation (GDPR) requirements when sharing data and require support with this.

“It would be good to have some more guidance on what we can and can’t do within the GDPR framework” Participant 5.

Around half of all areas used a multi-disciplinary team (MDT) to assess the suitability of their area for refugees and make decisions on acceptance. Membership of MDTs varied, most commonly we found a partnership between the local authority and CCG with direct input from primary care, education, social care, housing and the VCSE. An MDT was perceived to lessen the impact on health services through comprehensive planning of arrivals and resettlement with all partners ‘round the table’. Areas that had an MDT in place reported no issues with the HIS data flow process.

“Our approach in terms of multiagency approach is that the right people from the right organisations are making sure the doors are open so when the families arrive they are facilitated to register with that primary care practice, they’re not having to push open, the doors [are] already open” Participant 3.

#### 3.2.2. The MHA and Other Pre Departure Checks Can Be Out of Date on Arrival

There were concerns about the MHA and other health information being out of date on arrival. Some participants described a gap of up to 12 months between the MHA screening and arrival. In this time health conditions can worsen, there can be new issues and preganancies that create a unplanned impact on services.

“We have had problems with lack of information compared to what they find out in initial assessments upon arrival. One case where [an] individual had [a] kidney problem (on the MHA), the local authority questioned it and the health issue was described as ‘stable’ therefore the local authority took on family but upon arrival it was clear it was a complex issue. This is now placing a large burden on the local authority in which they were placed.” Participant 5.

One participant understood that the MHA was not an exact science.

“It’s a bit like taking your car in for a service, I can drive off the forecourt and four miles down the road the clutch blows.” Participant 3.

Two participants described complex cases where the pre-arrival checks did not pick out urgent health issues. This puts a burden on the resettlement providers and health services.

“I’ve been to A&E (Accident and Emergency) with clients on arrival day, I’ve had clients who are dehydrated, I’ve had clients who have not travelled with medication and I think there is a need for some better tick boxes prior to arrival that are very specific about this person.” Participant 1.

### 3.3. More Information Is Needed

The breadth of detail was considered insufficient by some participants as the MHA did not provide enough information to identify the wider needs that the resettlement providers require for safe resettlement. 

#### 3.3.1. Dental Health Needs Can Be High and There Are Barriers to Accessing NHS Dental Care

Some participants described that poor dental health was common with a need for urgent care on arrival.

“There is an overwhelming need for those in the resettlement scheme to have dental treatment, often urgent dental treatment. Very difficult to find an NHS dentist who will take them on.” Participant 6.

Accessing NHS primary care dentists was described as difficult in some areas. Two participants highlighted language barriers with one recognizing a need to provide funding to dentists to provide care to refugees.

“If something could be done to uniquely ringfence dental support.” Participant 6.

#### 3.3.2. Mental Health Screening and Access to Services Requires Improvement

Participants described refugees with unmet mental health needs and issues sourcing mental health support. Two participants recognised that this would require additional funding from the Home Office or CCG as these services would need additional capacity to support the needs of refugees. Two different areas had examples where they had used the existing resettlement monies to fund a specialist provider showing that funding of these services is possible. One participant described a reluctance amongst male refugees to engage with mental healthcare.

“Mental health has been the biggest challenge, access to services and getting males to access counselling as they have trauma. Moving people forward is one step forward two steps back as mental health impacts adherence to work and English language learning.” Participant 4.

#### 3.3.3. Special Educational Needs in Children (SEN) Screening and Access to Services Requires Improvement

A strong theme from most participants was the issue of child dependents arriving and entering the education system where SEN is then identified.

“We often we are going to have to scramble around during the first two, three weeks to find kids to find the right kind of support packages for children with disabilities and learning needs.” Participant 5.

Two rural areas described a lack of SEN education that was a barrier to resettlement.

“We are very cautious in relation to that, in terms of [a] child arriving with special education needs we might not be able to meet that need.” Participant 2.

Two participants spoke of issues where SEN was not recognised due to a perception that language was the barrier to learning.

“With schools the language barriers mean it takes a lot longer for children with SEN to be recognised.” Participant 7.

### 3.4. Resettlement Schemes Promote Refugee Health and Raises the Acceptance of Refugee Resettlement

Participants described how resettlement schemes had positive impacts on health and social care systems by funding services, increasing awareness and acceptability.

#### 3.4.1. Resettlement Schemes Funding Benefits the Wider Refugee Infrastructure

Two participants described how the resettlement scheme funding supports local areas’ refugee infrastructure and knowledge of refugee health. Two areas had been able to fund specific health services for refugees.

“We have a specialist adult mental health team for refugees, (as) people don’t exist in mainstream services that understand the refugee journey and the trauma and how trauma is different.” Participant 7.

#### 3.4.2. Resettlement Raises Acceptance of Refugees

Participants in rural areas new to resettlement described how refugee resettlement promoted civic and political engagement. This led to further resettlement through other resettlement schemes.

“I was able to cajole the local authority to say to their elected members this (refugee resettlement) would be a good thing to do because of a, b, c, d and we have done it elsewhere and it’s not as difficult as you think it is, so that was a good thing.” Participant 3.

Participants in areas that were new to resettlement described a positive outcome where refugee arrivals supported the wider VCSE and community sponsors to come together.

“One of the good things is it has brought community groups together.” Participant 3.

“The way we have seen rural resettlement work and grow in momentum, (with) movement around community sponsorship.” Participant 2.

## 4. Discussion

The findings of this study show that the HIS used for health data in refugee resettlement in England is fundamental for effective planning and supporting refugees’ health needs. Efficient and clear information sharing is an essential component of effective multi-agency processes [17]. Resettlement leads that used MDTs and information governance agreements had no data sharing issues and were able to better support refugees. Support from national agencies is needed with clear guidance to support data flows in those instances where robust processes are not in place.

Disruption or delay in the flow of data may limit the utility of MHA in the resettlement process. The findings in this study suggest that deficits in the information from the MHA are problematic and can lead to continuity of care issues that is reported in other studies [11]. The MHA meets its primary purpose, “to identify health conditions for which the treatment is required before travel to the UK” [8] but lacks sufficient information on mental health and child SEN needs. The public health impact of an incomplete or out of date HIS could lead to health protection risks (COVID-19 vaccination or testing status, Tuberculosis screening) [11], reduced readiness for school due to SEN, disrupted community integration, healthcare access due to unmet mental health needs and inappropriate housing due to health deterioration or recent pregnancies. These impacts can be reduced by improvements to resettlement policy and increasing the breadth and pace of international, national, and regional HIS.

Refugees are entitled to dental and mental healthcare through the NHS [18] and participants reported access issues and language barriers with these services. Similar access issues were also reported with local authority SEN assessments and services. In England this could be compounded by a significant shortage of NHS dentists and regional disparities in access [19], an overburdened mental health service [20] and a decline in SEN schools in England [21]. Consequently, due to the limited availability of, and access to these services, more pre-arrival information is required to enable effective planning forresettlement in areas that can meet the needs of the refugees. A call for more information in pre-arrival checks was also raised in the inspection of UK refugee resettlement schemes in 2020 by the UK Independent Chief Inspector of Borders and Immigration [5].

Despite allocated funding to authorities to support resettlement [22], there tends to be a focus in the literature on the strain that resettling refugees puts on the resources of the resettlement area [10,11]. Conversely, a strong theme of positivity emerged from the participants in this study. Resettlement funding supported the wider refugee infrastructure, effective inter-agency collaboration, and promoted further resettlement.

Encouragingly, some participants described how resettlement promoted civic and community engagement for resettling more refugees under community sponsorship programmes.

## 5. Strengths and Limitations

This is the only study assessing the HIS in England for refugee resettlement and participants represented a range of different local authority structures and experience of resettlement.

The small sample reduces generalizability. However, limited sample sizes are a well-accepted trade off associated with qualitative approaches but are well suited to probe into the independent thoughts of different groups [23].

Finally, the methods used may have introduced unidentified bias, such as researcher bias. Qualitative approaches to research make the elimination of researcher bias difficult [24]. Various techniques were used in this project to reduce this bias including reflexivity, piloting, double coding, and participant validation [24].

## 6. Conclusions

This study found that the HIS works in areas with defined information governance processes and an MDT. The main HIS health assessment tool; the MHA can lack up to date information and misses the wider health needs that are essential for resettlement.

Refugee resettlement had positive impacts for areas including joint working, seed funding and acceptance of refugees.

## Figures and Tables

**Figure 1 ijerph-18-10331-f001:**
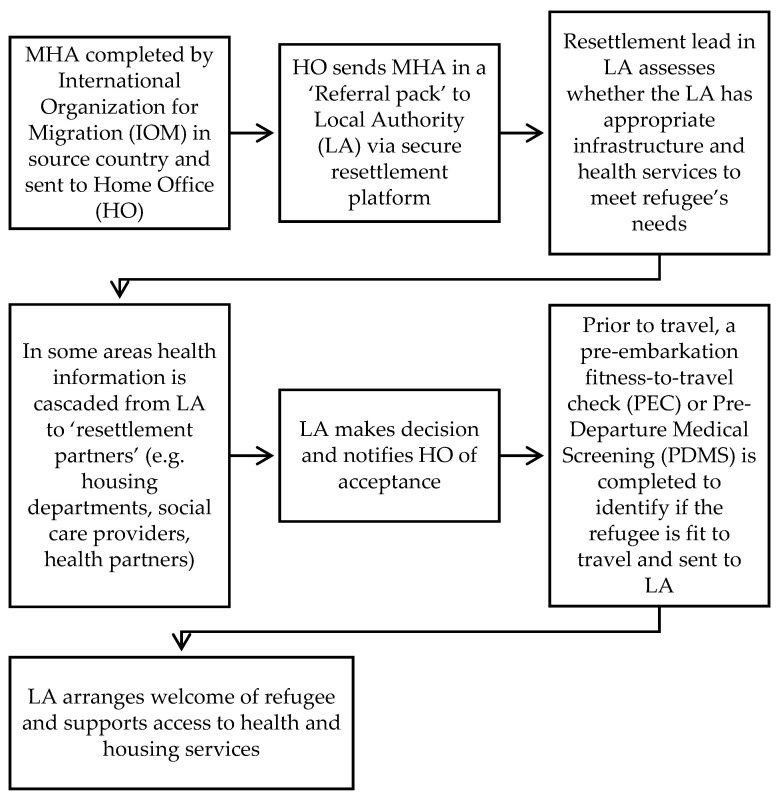
UK Medical Health Assessment (MHA) health information system.

**Table 1 ijerph-18-10331-t001:** UK Medical Health Assessment (MHA) aims and objectives.

	Aims and Objectives
**1**	To identify health conditions for which treatment is recommended before the individual travels to the UK, primarily for personal benefit, but also: to ensure the individual is settled in a location and accommodation that has appropriate facilities to meet their health and social care needs; to ensure current Tuberculosis (TB) screening practice is met in all pre-entry assessments;to identify and address conditions and diseases with public health significance before travel.
**2**	To offer immunisation, wherever possible, for the benefit of the individual and of society.
**3**	To assess the refugees’ fitness to travel to the UK and to their final destination within the UK.
**4**	To arrange special travel requirements (from a medical point of view) for the most vulnerable cases, when air travel might present additional risk to their health condition.
**5**	To identify and share medical information with the resettlement authorities in the UK for the purpose of organizing adequate care and treatment upon arrival in the UK.
Adapted from The Home Office Health protocol: Pre-entry health assessments for UK-bound refugees [Internet]. London: UK Government; 2020 [8]	

**Table 2 ijerph-18-10331-t002:** Characteristics of the participants.

	Participant 1	Participant 2	Participant 3	Participant 4	Participant 5	Participant 6	Participant 7
**Region**	North west	South West	East of England	South East	South East	South West	South West
**Sex**	Male	Female	Male	Female	Male	Male	Male
**Resettlement provider**	Lower tier	Upper tier	Upper tier	Upper tier	Lower tier	Upper tier	Lower tier
**Resettlement rate (resettlement numbers/total population size)**	0.0004	0.0003	0.0001	0.0002	0.0003	0.0004	0.0006
**Resettlement rate category**							
**High (>0.0004)**	X					X	X
**Medium (0.0003–0.0004)**		X			X		
**Low (<0.0002)**			X	X			

## Data Availability

Data is available from the corresponding author.

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
