# Peer review of "Service Evaluation of the English Refugee Health Information System: Considerations and Recommendations for Effective Resettlement"

_ijerph, 2021, doi:10.3390/ijerph181910331_

Round 1

Reviewer 1 Report

As the displacement of people from their home countries is a major global issue, the conditions of resettlement are of major importance in order to secure a decent future.

The manuscript gives a qualitative appraisal of the medical assessment process of refugees and the findings confirm what has already been shown in other refugee incoming countries.

It would be greatly benefit the paper to include some additional information on the resettlement process in order to allow the reader to better understand the findings of the study.

More specifically, additional information on the following would be very helpful:

Please provide some information on the numbers of refugees accepted by other countries in the European area, especially in the south of the region (Italy, Greece, Spain etc), to give a clearer picture of the displaced population in the recent years.

Short description of the legal framework of refugee re-settlement in England? Are specific groups prioritized eg families, country of origin, age?

After resettlement, is there a period of monitoring adjustment to the new living conditions? Are the refugees being re-assessed in England and if so for how long? What is their residence status?

How does the whole resettlement scheme work? The refugees apply for resettlement to England from their home country or how?

Who are members of the multi-disciplinary team that assesses the area resettlement suitability of the refugees?

Are there English language course offered to the refugees after arrival?

Do the refugees have access to their MHA information?

Thank you.

Author Response

Dear reviewer 1. 

Thank you for providing notes and comments. We agree with all your points and have addressed these in the following section. The additions are are highlighted in the manuscript text in yellow

As the displacement of people from their home countries is a major global issue, the conditions of resettlement are of major importance in order to secure a decent future.

We agree, and feel that relatively simple changes to host countries' processes and systems could benefit the refugees in the short term (resettlement) and longer term (integration). 

The manuscript gives a qualitative appraisal of the medical assessment process of refugees and the findings confirm what has already been shown in other refugee incoming countries.

We agree, through some initial conversations with the IOM, resettlement providers, the Home Office and from the limited literature we were aware there were issues. We used a qualitative approach to create evidence of the issues and feel that there will be national/international commonalties. 

It would be greatly benefit the paper to include some additional information on the resettlement process in order to allow the reader to better understand the findings of the study.

Thank you for your comments, they were helpful in supporting us to clearly define the context and process of the refugee resettlement program, as we focussed on the health information system too early without sufficient context building. We agree with all your comments, and have actioned these. Please see below for the responses to your questions. 

  • Please provide some information on the numbers of refugees accepted by other countries in the European area, especially in the south of the region (Italy, Greece, Spain etc), to give a clearer picture of the displaced population in the recent years.
  • We have added a line that describes locations for the resettlement of refugees in Europe. 
  • Lines 38-41  In 2019, the vast majority of displaced people that applied for resettlement in Europe were settled in Germany (9640), Sweden (5408), Norway (3507), the UK (3507), and France (3311), with fewer settled in the southern region of Europe, such as Spain (1193), Italy (413) and Portugal (879). 
  • Short description of the legal framework of refugee re-settlement in England? Are specific groups prioritized eg families, country of origin, age?
  • The UK’s legal framework for categorising refugees is similar to the UNHCR framework that is quoted in the introduction. We have added a line on prioritization for the UK schemes. 
  • Line 43-45. Resettlement priority for UK is based on the urgency of security and/or medical needs with priority to women and children that are risk of violence or exploitation and refugees with family links in the UK
  • After resettlement, is there a period of monitoring adjustment to the new living conditions? Are the refugees being re-assessed in England and if so for how long? What is their residence status?
  • We will add a line to the introduction that summarises the follow up of the refugee, we did not ask about this process during the interviews, but the process is a local authority function. This is an interesting question, and I think from our research that some areas would likely have good follow up or support through their systems/VCSE partners, others less so. However, this is conjecture and not evaluated in our study. 
  • Line 63-65 Monitoring and evaluation of refugee integration is a local authority responsibility as the Home Office has little contact with the refugees post-arrival 
    • There is some variation between different UK resettlement schemes. Resettlement is a 5 year process in all. Five years Leave to Remain (LTR) to UK resettlement scheme and is replicated in the vulnerable persons and vulnerable resettlement schemes. The Gateway scheme grants Indefinite Leave to Remain (ILR). We feel it’s beyond the scope (and word count) to explain these, so we have added a line to describe the residency offer. 
  • Line 46-48. Most schemes offer Leave to Remain residency for five years, with Gateway providing the Indefinite Leave to Remain.  
  • How does the whole resettlement scheme work? The refugees apply for resettlement to England from their home country or how?
  • This is a useful comment as we have not set the context for the wider resettlement process. Potential cases are identified by the United Nations Commissioner for Refugees (UNHCR) and are then referred to the UK Home Office who verify them against eligibility criteria.  We have added a line on the process
  • Line 47-49 The UNHCR provides an out of country assessment for each application to determine their eligibility and refugee status. Cases eligible for the UK resettlement schemes are forwarded to the Home Office who determine if the refugee can be resettled 
  • Who are members of the multi-disciplinary team that assesses the area resettlement suitability of the refugees?
  • This is a useful point to clarify, as we do not do this in the text. As the VPRS process varies from provider to provider, membership of multi-disciplinary teams utilised by some providers also varies. In some instances there aren’t MDTs.  Most commonly we found a partnership between the local authority and medical providers. This could be via a CCG contact or direct input from GPs. Other team members include representatives from education providers, social care, mental health services and community partners. Membership may also vary depending on the situation of the refugee family, for example having child caseworkers involved with families with young children. We have added a few lines to summarise these. 
  • Line 200-202  Membership of MDTs varied, most commonly we found a partnership between the local authority and CCGs with direct input from GPs, education, social care, housing and the VCSE. 
  • Are there English language courses offered to the refugees after arrival?
  • Yes, provision of English language services is decided by the resettlement provider, we have added information and clarity about the role of the local authority. English language courses are offered to people on the resettlement schemes. We did not identify which organisation provides the support, in some cases we think it is the local authority commissioning a service, or in other case the VCSE provide the support and provide a service.  
  • Line 58-59 The local authority will arrange housing, education, and adult English language lessons.
  • Do the refugees have access to their MHA information?
  • Thank you for picking this up, this is an important point. We have added a line to state that refugees do have access to a hardcopy of their MHA. 
  • Line 66. The refugees will also receive a copy of their MHA for their records. 

Thank you for your time and supportive guidance. 

Kind regards

The authors. 

Reviewer 2 Report

With interest, I have read the ms “Service evaluation of the English refugee health information system: considerations and recommendations for effective resettlement”, by Dunn and colleagues. The article is well-written and thought-provoking.

I have just a few minor comments/suggestions.

Overall, I’d like to see, in Discussion/Conclusion) more on the Public Health impact of the presented study and possible future implications from a policy perspective. Not only in England/UK, but also as a possible approach for a change of pace in migrant and refugee health.

Materials and Methods.

Line 117 – “A question guide was created with a Consultant in Public Health and a migrant health researcher.” If both the PH consultant and researcher are listed as author of the study, it should be very informative to add their initials (in round brackets).

Line 133 (and Data analyses paragraph) – Similarly, I understand that TD, AB, and SH are authors’ initials, but they seem author-invented acronymous. I suggest to report them in round brackets, as per standard publication guidelines.

In Appendix B, coding and theming trees present an important part of this research but it’s very difficult to read: I suggest to present them on two separate landscape (horizontal) orientated pages.

Author Response

Dear reviewer 2. 

Thank you for providing notes and comments. We agree with all your points and have addressed these in the following section. The additions are highlighted in the manuscript text in turquoise. 

With interest, I have read the ms “Service evaluation of the English refugee health information system: considerations and recommendations for effective resettlement”, by Dunn and colleagues. The article is well-written and thought-provoking.

  • Thank you for these comments, it has been a collaborative effort and we are glad it was thought provoking. This was the aim of our write up of the service evaluation as we felt there were interesting and useful findings. 

I have just a few minor comments/suggestions.

Overall, I’d like to see, in Discussion/Conclusion) more on the Public Health impact of the presented study and possible future implications from a policy perspective. Not only in England/UK, but also as a possible approach for a change of pace in migrant and refugee health.

  • Thank you, we have added some lines on the public health impact on policy,  this is an important omission you’ve picked up and something that we discussed when planning the paper
  • Lines 311-317 incorporates the new comments-  The public health impact of an incomplete or out of date HIS could lead to health protection risks (COVID-19 vaccination, or testing status, Tuberculosis screening), reduced readiness for school due to SEN, disrupted community integration and healthcare access due to unmet mental health needs and inappropriate housing due to health deterioration or recent pregnancies. These impacts can be reduced by improvements to resettlement policy and increasing the pace of international, national, and regional HIS. 

Materials and Methods.

Line 117 – “A question guide was created with a Consultant in Public Health and a migrant health researcher.” If both the PH consultant and researcher are listed as author of the study, it should be very informative to add their initials (in round brackets).

  • Actioned, we have added the initials for these two authors. 

Line 133 (and Data analyses paragraph) – Similarly, I understand that TD, AB, and SH are authors’ initials, but they seem author-invented acronymous. I suggest to report them in round brackets, as per standard publication guidelines.

  • Actioned, we have clarified roles, and added initials in round brackets. 

In Appendix B, coding and theming trees present an important part of this research but it’s very difficult to read: I suggest to present them on two separate landscape (horizontal) orientated pages.

  • Thank you, we will work with the editors if successful to action this as the formatting did not enable this. 

Thank you for your time and supportive guidance. 

Kind regards

The authors.